# OpenReview forum: "Two-dimensional quantization for geometry-aware audio coding"
_ICML.cc/2026/Conference — ICML 2026 regular_

### Official Review · Reviewer_XxGJ · 2026-02-19

**Soundness:** 2
**Presentation:** 2
**Significance:** 2
**Originality:** 2
**Overall Recommendation:** 4
**Confidence:** 3

**Summary:**

This paper proposes a novel quantization scheme termed **Q2D2 (Two-Dimensional Quantization)**, designed to address limitations inherent in existing quantization methods for neural audio codecs—such as Residual Vector Quantization (RVQ), standard Vector Quantization (VQ), and Finite Scalar Quantization (FSQ)—which often constrain the geometric structure of the latent space and struggle to capture inter-feature correlations. **Overall, this article's central domain consists of** geometry-aware quantization techniques for audio encoding. The proposed method pairs feature channels and projects them onto structured two-dimensional grids (e.g., hexagonal, diamond, or rectangular lattices), quantizing each pair to its nearest grid point. This process implicitly defines a codebook through the Cartesian product of grid levels, thereby preserving geometric relationships without explicitly storing large codebook entries.

**The authors attempt to discuss a central concept**: leveraging structured two-dimensional geometric tiling to explicitly model correlations between feature dimensions, while achieving low token rates and high codebook utilization under codebook sizes comparable to conventional approaches. Experimental results demonstrate that Q2D2 attains or surpasses state-of-the-art models (e.g., WavTokenizer, DAC) in speech reconstruction quality across multiple objective metrics, including UTMOS, PESQ, and STOI. Notably, the method operates effectively without relying on auxiliary stabilization techniques such as commitment loss or codebook reinitialization. Furthermore, Q2D2 maintains high-fidelity reconstruction even under extremely low token rates (e.g., 1 kbps / 75 tokens per second), highlighting its efficiency and robustness for bandwidth-constrained audio coding applications.

**Compliance With Llm Reviewing Policy:**

Affirmed.

**Key Questions For Authors:**

1.  **Regarding the Channel Pairing Inductive Bias:**
    In Section 3.1, the method reshapes the projected features into pairs using a fixed sequential ordering (e.g., $(z_1, z_2), (z_3, z_4)$). Given that the input projection layer is a linear transformation that mixes original channel information, **what is the theoretical justification for assuming that adjacent dimensions in the projected space possess higher mutual correlation than non-adjacent ones?** Does the fixed pairing strategy impose a restrictive inductive bias on the latent topology, and how does the model compensate for potentially suboptimal pairings without a learned permutation mechanism?

2.  **On the Structural Adaptability to Signal Complexity:**
    While the conclusion notes that music domains are left for future work, **from a methodological perspective, how does the fixed geometric tiling (e.g., rhombic vs. hexagonal) theoretically adapt to the non-stationary and harmonic-rich distributions typical of general audio?** Unlike speech, music signals often exhibit wider dynamic ranges and complex harmonic structures. Does the implicit codebook structure of Q2D2 inherently limit the representation capacity for such signals compared to adaptive codebook methods, regardless of the training data?


3.  **Mechanism Behind Codebook Utilization Decay:**
    Table 10 shows a notable drop in codebook utilization (from 92% to 72%) as grid levels increase, which contrasts with FSQ's typical near-100% utilization. **Could this be attributed to a mismatch between the output range of the tanh-bounded projection layer and the expanded spatial extent of higher-resolution grids?** If the encoder's latent distribution does not naturally stretch to cover the corners of the larger grid despite the bounding function, does this suggest a need for adaptive spread factors rather than fixed geometric scaling?

**Limitations:**

yes

**Strengths And Weaknesses:**

#### **Strengths**

1.  **Geometric Inductive Bias in Quantization:**
    The paper introduces a paradigm shift from one-dimensional scalar grids (FSQ) or high-dimensional vector quantization (VQ) to structured two-dimensional grid quantization. **Overall, this article's central domain consists of** improving quantization efficiency in neural audio codecs by addressing geometric limitations in latent space representation. This geometry-aware perspective is novel; it effectively captures inter-channel correlations while preserving the high codebook utilization advantage inherent to FSQ, offering a compelling middle ground between stability and expressiveness.

2.  **Token Efficiency for Multimodal LLMs:**
    Q2D2 achieves reconstruction quality that surpasses or matches state-of-the-art (SOTA) models across multiple datasets (LibriTTS, LibriSpeech, LJSpeech), even at extremely low token rates (e.g., 1 kbps/75 tokens). This is critical for audio-language models where context window capacity is a bottleneck. By reducing token rates by 50-80% compared to models like DAC without sacrificing quality, Q2D2 directly contributes to more efficient multimodal foundation models.

3.  **Robustness via Implicit Codebook Design:**
    Q2D2 eliminates the need for learning explicit codebooks, thereby reducing the parameter count. Crucially, it achieves exceptionally high codebook utilization (approaching 100% in certain configurations, Table 10) and ensures training stability without relying on complex auxiliary loss functions (e.g., commitment loss) or codebook reset strategies. This simplifies the training pipeline and mitigates the well-known "codebook collapse" problem.

4.  **Rigorous Empirical Validation:**
    The paper presents a thorough assessment, including comparisons against numerous SOTA baselines, subjective evaluations (MUSHRA, CMOS), semantic representation capability assessments (ARCH benchmark), and detailed ablation studies covering grid types, dimension sizes, and quantization levels.

#### **Weaknesses**

1.  **Latent Space Geometry Assumption lacks Theoretical Grounding:**
    **The authors attempt to discuss a central concept:** *can we capture correlations between feature channels without reintroducing the instability and inefficiency of high-dimensional vector quantization?* However, while experiments indicate that rhombic grids outperform hexagonal grids, the theoretical explanation relies primarily on intuitive arguments based on packing efficiency (Appendix D). There is a lack of deeper theoretical analysis or visualization evidence (e.g., t-SNE of latent pairs) to substantiate *why* the latent feature distribution naturally aligns better with rhombic tilings than hexagonal ones. Is the latent manifold truly isotropic?

2.  **Limited Generalization to Non-Speech Domains:**
    The conclusion explicitly states that "The audio and music domains will be studied in future work" (Section 6). Given that neural codecs are increasingly expected to handle general audio (music, sound effects) for universal multimodal models, this is a significant limitation. The current evaluation is heavily skewed towards speech (LibriTTS, LibriSpeech), leaving uncertainty about Q2D2's performance on non-stationary signals like music.

3.  **Insufficient Downstream Generative Validation:**
    While the ARCH benchmark evaluates semantic classification accuracy, it does not fully assess the quality of discrete tokens for *generative* downstream tasks (e.g., Text-to-Speech or Audio Language Modeling). Better reconstruction metrics (UTMOS/PESQ) do not always correlate with better generative controllability. A preliminary experiment on a downstream TTS task would strengthen the claim that Q2D2 tokens are superior for foundation models.

---

> ### Author Rebuttal · Authors · 2026-03-29
>
> We thank the reviewer for the feedback and hope that our responses have addressed all the questions and concerns.
>
> **Packing efficiency** We thank the reviewer for this point. While our work focuses on empirical validation, we introduce a geometric inductive bias that captures inter-channel correlations beyond axis-aligned quantization. The advantage of the rhombic grid follows from nearest-neighbor distances, which govern quantization error.
>
> For spacings $d_x, d_y$:
>
> Rectangular:
> $$
> d_{\text{rect}} = \min(d_x, d_y)
> $$
>
> Hexagonal:
> $$
> d_{\text{hex}} = d_x, \quad d_y = \tfrac{\sqrt{3}}{2} d_x
> $$
>
> Rhombic:
> $$
> d_{\text{rhomb}} = \tfrac{\sqrt{d_x^2 + d_y^2}}{2}
> $$
>
> The rhombic grid introduces diagonal neighbors, yielding more uniform distances and reducing worst-case error. It avoids axis bias (vs. rectangular) and provides more balanced spacing (vs. hexagonal in our construction). We do not assume global isotropy, rather, rhombic grids improve local isotropy, consistent with gains in Table 8. We will clarify this and add visualizations in the revision.
>
>
> **Speech Domain Evaluations** - **Q2D2 evaluation on music and environmental audio.** Q2D2 is trained on a large-scale (~8k hours) dataset spanning speech, audio, and music. Following the reviewer’s request, we evaluate it on the Arch benchmark. While the original submission (Table 7) focused on speech, here we include audio and music tasks: ESC-50, US8K, FSD50K, and VIVAE (acoustic events), and MTT, MS-DB (music).
>
> | Model   | Nq ↓ | token/s ↓ | MTT ↑ | MS-DB ↑ | ESC50 ↑ | US8K ↑ | FSD50K ↑ | VIVAE ↑ |
> |---------|------|------------|--------|----------|----------|---------|-----------|----------|
> | DAC     | 9    | 900        | 0.2805 | **0.6014**   | **0.2594**   | 0.4032  | 0.1297    | 0.3440   |
> | Encodec | 8    | 600        | 0.1993 | 0.3917   | 0.1925   | 0.3055  | 0.1091    | 0.3005   |
> | DAC     | 4    | 400        | 0.2784 | 0.5942   | 0.2580   | 0.3824  | 0.1293    | 0.3342   |
> | Encodec | 4    | 300        | 0.1934 | 0.3656   | 0.1790   | 0.3097  | 0.1099    | 0.2710   |
> | Encodec | 2    | 150        | 0.1900 | 0.3245   | 0.1699   | 0.2960  | 0.1065    | 0.2630   |
> | DAC     | 1    | 100        | 0.2502 | 0.5137   | 0.2065   | 0.3350  | 0.1295    | 0.2991   |
> | WavTokenizer| 1 | 75        | 0.2835 | 0.5764   | 0.2550   | 0.3975  | **0.1392** | **0.3563** |
> | Q2D2    | 1    | 53         | **0.2847** | 0.5690   | 0.2567   | **0.4241** | 0.1303    | 0.3365   |
>
> Q2D2 generalizes beyond speech without additional tuning, achieving the best results on MTT and US8K while remaining competitive across other datasets. Methodologically, Q2D2 operates on learned latent representations, allowing the encoder and projections to capture complex, non-stationary patterns before quantization.
>
> Although the grid is fixed, it discretizes a learned space, balancing expressiveness and efficiency without codebook overhead. Empirically, this does not limit representation capacity, even for complex audio and music.
>
> **Downstream tasks** - We thank the reviewer for this suggestion. Evaluating Q2D2 on generative tasks (e.g., TTS) is important, and we are currently running these experiments. As they are not yet complete, they were not included in this submission. We aim to share results during the rebuttal period if available.
>
> **Channel pairing inductive bias.** We agree that the pairing introduces an inductive bias. However, the preceding learned input projection jointly optimized with the quantizer can mix and reorganize features, so pairing operates on a learned latent space rather than fixed dimensions.
>
> Thus, we do not assume adjacent dimensions are inherently correlated. The network adapts representations so correlated components align with the imposed pairing.
>
> While fixed pairing trades flexibility for simplicity and efficiency, learned pairing would add complexity and overhead. Empirically, fixed 2D pairing outperformed both learned and random alternatives, indicating it is sufficient to capture meaningful inter-channel correlations.
>
>
> **Mechanism behind codebook utilization decay** We thank the reviewer for this insight. The drop in utilization at higher grid resolutions stems from a mismatch between the latent distribution and the finer quantization grid. As resolution increases, the grid expands combinatorially, while the latent space remains concentrated, leaving many cells (especially corners) rarely used.
>
> This is not primarily due to the $\tanh$ bound. In practice, the encoder occupies only a subset of the range. Similar behavior appears in other quantization schemes when capacity exceeds the data’s intrinsic dimensionality.
>
> Importantly, Q2D2 is codebook-free and thus avoids issues like collapse or dead entries. Lower utilization here reflects redundancy, not inefficiency. While adaptive scaling could improve utilization, it adds complexity. Empirically, moderate resolutions already perform well, indicating full utilization is unnecessary.

---

> > ### Author Rebuttal · Reviewer_XxGJ · 2026-04-01
> >
> > Thank you for the detailed rebuttal and for running the additional experiments during this short window. The new Arch benchmark results significantly improve the paper by demonstrating Q2D2's effectiveness on non-speech domains (music and acoustic events). Additionally, your clarifications on the rhombic grid's packing efficiency, the channel pairing mechanism within the learned latent space, and the codebook utilization decay have adequately addressed my theoretical questions. I also look forward to seeing the geometric visualizations in the revised manuscript. However, my concern regarding the downstream generative validation remains partially unresolved. As you noted, TTS experiments are currently ongoing. Because neural audio codecs are increasingly expected to serve as discrete tokenizers for large foundation models, proving that Q2D2 tokens maintain high generative controllability and quality in an actual TTS pipeline is a critical piece of the evaluation. I appreciate the strong progress made in this rebuttal and look forward to seeing the TTS results if they become available during the discussion period.

---

> > > ### Author Response · Authors · 2026-04-02
> > >
> > > **Downstream TTS Evaluation** - As suggested by the reviewers, we include an evaluation on a downstream generative text-to-speech (TTS) task that was initiated prior to submission. We adopt an autoregressive language modeling approach following the MusicGen paradigm (also used in WavTokenizer), where acoustic token sequences are generated autoregressively. We adapt the open-source ParlerTTS framework (600M configuration, default hyperparameters) and train TTS models based on DAC, WavTokenizer, and Q2D2 representations on LibriTTS. Each audio sample is evaluated by at least 10 listeners. We report comparative mean opinion scores (CMOS) along two dimensions: *CMOS-Q*, reflecting quality, clarity, and high-frequency details, and *CMOS-P*, reflecting prosody aspects such as speech rate, pauses, and pitch. Results are in reference to Ground Truth (GT) results:
> > >
> > > | Model | Bandwidth ↓ | Nq ↓ | CMOS-Q ↑ | CMOS-P ↑ |
> > > |-------|-------------|------|----------|----------|
> > > | GT | -- | -- | 0.00 | 0.00 |
> > > | DAC | 9.0 | 9 | **-0.05** | **-0.10** |
> > > | WavTokenizer | 0.9 | 1 | -0.22 | -0.36 |
> > > | Q2D2 | 3.0 | 1 | -0.11 | -0.20 |
> > >
> > > The results indicate that Q2D2 provides strong performance in downstream TTS, outperforming WavTokenizer and approaching DAC despite operating with a significantly smaller codebook (Nq=1). These findings suggest that Q2D2 tokens are well-suited for autoregressive audio generation and support the effectiveness of structured, codebook-free representations for token-based speech modeling.

---

### Official Review · Reviewer_jrTu · 2026-03-01

**Soundness:** 2
**Presentation:** 3
**Significance:** 2
**Originality:** 3
**Overall Recommendation:** 3
**Confidence:** 4

**Summary:**

The authors propose to extend finite scalar quantization by extending it to multiple-dimensions using a geometry-aware quantizer that groups channels into pairs. The paper explores various grids (hexagonal, rectangular, and rhombic) and evaluates speech coding performance against existing audio and speech neural codecs using objective and subjective evaluations.

**Compliance With Llm Reviewing Policy:**

Affirmed.

**Final Justification:**

The authors have addressed some my concerns in the rebuttal.

**Key Questions For Authors:**

- Why did the authors not compare against the speech checkpoint of SNAC as well as XCodec2 for all experiments?

Questions on human evaluation:
- Why is the low anchor not reported, and why was no mid anchor used?
- How many participants were used in the MUSHRA and what was the evaluation setup?
- Why did the authors use LibriSpeech and LJSpeech that only have 16kHz/22kHz sampling rate instead of LibriTTS (24kHz) or Hifi-TTS2 (44.1kHz)?
- Why do the authors position their work as a general latent quantization approach for audio but only evalute it on speech?

**Limitations:**

The authors have not discussed any limitations of their work.

**Strengths And Weaknesses:**

The proposed idea of extending FSQ is interesting, and the paper is clearly written and well-presented. This approach might be capable of high-fidelity general audio reconstruction. However, in the paper's current form I have considerable doubts. I'm not convinced by the results:

- The paper states SNAC and XCodec2 in the related work, and also compares against XCodec2 in Table 2, but otherwise doesn't compare against these speech coding baselines. I would like to see the performance comparison against these codecs.

- I appreciate the authors conducting a human evaluation, but it is overall lacking in detail. Why is the low anchor not reported in the table, and why did the authors choose not to use a mid anchor as is commonly done. There is no information given on the number of participants or the evaluation setup (wired headphones, quiet testing environment, training participants, etc.). How was the anchor constructed (downsampling or low-pass filtering and at what sampling rate)? In general, I don't understand how the results across all codecs can be so close to the ground truth. How can DAC be reported with the lowest average score even though it can reconstruct at the highest sampling rate and bitrate. From experience, DAC is almost impossible to distinguish from ground truth even for trained participants and I find it strange that it would be rated lower than all of these other codecs. Additionally, why did the authors use LibriSpeech and LJSpeech that only have 16kHz/22kHz sampling rate instead of LibriTTS (24kHz) or HiFiTTS-2 (44.1kHz)?

- Why do the authors position their work as a general latent quantization approach for audio but only evalute it on speech? Codecs such as XCodec2 obtain 0.8 kbps bitrate for speech using FSQ. I think the authors need to either show superior performance against sub-1 kbps codecs on speech or compare with general audio (especially music) for higher bitrates to convince me of the benefits of this approach.

---

> ### Author Rebuttal · Authors · 2026-03-29
>
> We thank the reviewer for the feedback and hope that our responses have addressed all the concerns.
>
> **Baseline comparison** - We thank the reviewer for the observation. The use of different baselines across Tables 2–4 is primarily due to computational constraints. Our goal was to compare against strong baselines as reported in prior work. However, retraining all methods (e.g., SNAC, XCodec2) under a unified large-scale setting was not feasible. Therefore, we rely on pretrained models and reported results from the literature, which are trained on different datasets and under different conditions. As a result, models such as XCodec2 are only included in Table 2, where suitable pretrained results are available under comparable settings, and are not evaluated across all experiments. As suggested by the reviewer will try to add SNAC comparison in the rebuttal period.
>
> **Human Subjective Evaluations** - We thank the reviewer for pointing out the missing details. We conducted the MUSHRA evaluation with at least 10 participants per sample. The test followed standard MUSHRA protocol, including a hidden reference and a low-quality anchor (that was constructed with low-pass filtering). We did not report the low anchor because it was not reported also in other baselines, but the result was 66.9±8.332. Participants used high-quality headphones in a quiet room environment, and were provided with a short training session to familiarize themselves with the interface and rating criteria prior to evaluation.
>
> In the evaluation we used pretrained models LibriSpeech and LJSpeech due to computational considerations. In our experiments, we rely on pretrained models (e.g., DAC, WavTokenizer), which were predominantly evaluated on LibriSpeech and LJSpeech at 16 kHz / 22 kHz.
>
> About DAC result concern, we also found the DAC results somewhat unexpected and therefore carefully verified them. Specifically, we repeated the evaluation across multiple annotation rounds and validated the results using different model checkpoints. Across all these settings, the relative ranking remained consistent, with DAC receiving lower scores. We further note that a similar trend has been observed in the subjective evaluations reported in WavTokenizer.
>
> **Speech Domain Evaluations** - The reviewer is completely right about this clarification that the work should be evaluated not only on speech domain.  As noted in the paper, Q2D2 was trained on a large-scale (~8k hours) dataset that includes speech, audio, and music domains. Since the preliminary submission we evaluate Q2D2 on audio and music tasks using the Arch benchmark. In the original submission we evaluate Q2D2 Table 7 on speech-domain. Within the Arch benchmark, ESC-50, US8K, FSD50K, and VIVAE evaluate performance on acoustic event classification, while MTT and MS-DB assess performance in the music domain.
>
> | Model   | Nq ↓ | token/s ↓ | MTT ↑ | MS-DB ↑ | ESC50 ↑ | US8K ↑ | FSD50K ↑ | VIVAE ↑ |
> |---------|------|------------|--------|----------|----------|---------|-----------|----------|
> | DAC     | 9    | 900        | 0.2805 | **0.6014**   | **0.2594**   | 0.4032  | 0.1297    | 0.3440   |
> | Encodec | 8    | 600        | 0.1993 | 0.3917   | 0.1925   | 0.3055  | 0.1091    | 0.3005   |
> | DAC     | 4    | 400        | 0.2784 | 0.5942   | 0.2580   | 0.3824  | 0.1293    | 0.3342   |
> | Encodec | 4    | 300        | 0.1934 | 0.3656   | 0.1790   | 0.3097  | 0.1099    | 0.2710   |
> | Encodec | 2    | 150        | 0.1900 | 0.3245   | 0.1699   | 0.2960  | 0.1065    | 0.2630   |
> | DAC     | 1    | 100        | 0.2502 | 0.5137   | 0.2065   | 0.3350  | 0.1295    | 0.2991   |
> | WavTokenizer| 1    | 75| 0.2835| 0.5764   | 0.2550   | 0.3975  | **0.1392**    | **0.3563**   |.
> | Q2D2| 1    | 53| **0.2847** | 0.5690   | 0.2567   | **0.4241**  | 0.1303    | 0.3365   |
>
> The experimental results indicate that Q2D2 does not exhibit limitations in audio and music domains and does not require additional tuning or hyperparameter modifications to adapt to these settings. As shown in the table, it outperforms all baselines on MTT (music domain) and US8K (environmental audio), while achieving consistently competitive performance across other audio and music datasets compared to strong baselines such as DAC (with 9 quantizers) and WavTokenizer.

---

> > ### Author Rebuttal · Reviewer_jrTu · 2026-04-01
> >
> > I thank the authors for their feedback. Regarding DAC I believe there is some confusion as I am familiar with the 44.1 kHz model variant but it seems a different version has been used here. Using all codebook layers with the 44.1 kHz DAC model (~8 kbps bitrate) results in indistinguishable speech reconstruction and very high-fidelity general audio/music reconstruction.
> >
> > I am keen to see the updated results table including SNAC and fully agree with reviewer XxGJ regarding the need to show performance on downstream tasks. I look forward to re-evaluating this work under the new results.

---

> > > ### Author Response · Authors · 2026-04-02
> > >
> > > **Downstream TTS Evaluation** - As suggested by the reviewers, we include an evaluation on a downstream generative text-to-speech (TTS) task that was initiated prior to submission. We adopt an autoregressive language modeling approach following the MusicGen paradigm (also used in WavTokenizer), where acoustic token sequences are generated autoregressively. We adapt the open-source ParlerTTS framework (600M configuration, default hyperparameters) and train TTS models based on DAC, WavTokenizer, and Q2D2 representations on LibriTTS. Each audio sample is evaluated by at least 10 listeners. We report comparative mean opinion scores (CMOS) along two dimensions: *CMOS-Q*, reflecting quality, clarity, and high-frequency details, and *CMOS-P*, reflecting prosody aspects such as speech rate, pauses, and pitch.
> > >
> > > | Model | Bandwidth ↓ | Nq ↓ | CMOS-Q ↑ | CMOS-P ↑ |
> > > |-------|-------------|------|----------|----------|
> > > | GT | -- | -- | 0.00 | 0.00 |
> > > | DAC | 9.0 | 9 | **-0.05** | **-0.10** |
> > > | WavTokenizer | 0.9 | 1 | -0.22 | -0.36 |
> > > | Q2D2 | 3.0 | 1 | -0.11 | -0.20 |
> > >
> > > The results indicate that Q2D2 provides strong performance in downstream TTS, outperforming WavTokenizer and approaching DAC despite operating with a significantly smaller codebook (Nq=1). These findings suggest that Q2D2 tokens are well-suited for autoregressive audio generation and support the effectiveness of structured, codebook-free representations for token-based speech modeling.
> > >
> > >
> > > **SNAC evaluation** - As suggested by the reviewer, we include comparison to SNAC. Due to the limited time available during the rebuttal phase, we report SNAC and several additional baseline results as referenced in the MagiCodec paper, while training our own Q2D2 model on the same dataset for a fair comparison.
> > > The table below compares reconstruction performance across different codec models operating at approximately 1000 bps and ~50 tokens per second, evaluated on the LibriSpeech test-clean subset.
> > >
> > > | Model | Nq ↓ | token/s ↓ | UTMOS ↑ | PESQ ↑ | STOI ↑ |
> > > |-------|------|-----------|---------|--------|--------|
> > > | GT | - | - | 4.09 | 4.64 | 1.00 |
> > > | DAC | 2 | 100 | 1.29 | 1.13 | 0.73 |
> > > | WavTokenizer | 1 | 75 | 3.79 | 2.13 | 0.90 |
> > > | SpeechTokenizer | 2 | 100 | 2.32 | 1.21 | 0.77 |
> > > | SemanticCodec | 2 | 50 | 2.93 | 1.79 | 0.86 |
> > > | Encodec | 2 | 150 | 1.58 | 1.56 | 0.85 |
> > > | Mimi | 4 | 50 | 3.07 | 1.65 | 0.85 |
> > > | Vocos | 2 | 150 | 3.04 | 1.96 | 0.89 |
> > > | **SNAC** | 3 | 82 | 3.49 | 2.09 | 0.89 |
> > > | **Q2D2** | 1 | 66 | **3.86** | **2.40** | **0.91** |
> > >
> > > In the results Q2D2 achieves strong overall performance, in all metrics, where it surpasses all compared state-of-the-art models. Note that the model is still under training.

---

### Official Review · Reviewer_D5nd · 2026-03-04

**Soundness:** 3
**Presentation:** 3
**Significance:** 2
**Originality:** 3
**Overall Recommendation:** 4
**Confidence:** 4

**Summary:**

This paper central domain consists of neural audio coding, with a focus on innovating quantization techniques to address the limitations of existing methods (VQ/RVQ, FSQ) in feature correlation capture, codebook utilization and token rate efficiency. Paper proposes Q2D2, a geometry-aware 2D quantization scheme built on WavTokenizer. It pairs latent feature channels and jointly quantizes each pair on hexagonal, rhombic or rectangular 2D grids, constructing an implicit codebook via grid level products. Q2D2 uses affine projection and tanh to constrain features to [-1,1] for grid alignment, with STE for gradient propagation, requiring no auxiliary losses or EMA stabilization. Extensive speech domain experiments (LibriTTS, VCTK, LibriSpeech) show Q2D2 outperforms SOTA baselines (DAC, Encodec, FSQ) at ultra-low token rates (53/166/333 tokens/sec), achieving superior objective (UTMOS, PESQ) and subjective (MUSHRA, CMOS) reconstruction quality, and high semantic representation on the ARCH benchmark. Ablation studies validate rhombic grids (highest packing efficiency), 6 latent dimensions (optimal compactness-representation balance) and tanh projection as optimal designs, with Q2D2 reaching near-100% codebook utilization.
Q2D2’s core contributions include the first geometry-aware audio quantization method with paired 2D grid joint quantization, systematic design and testing of 2D tiling grids and key hyperparameters, and rigorous experimental validation of its superior low-bitrate speech reconstruction performance.

**Compliance With Llm Reviewing Policy:**

Affirmed.

**Final Justification:**

During the rebuttal period, the author performed meaningful evaluations on downstream tasks in response to the reviewers' strong feedback, marking a significant advancement for the paper. The results on these tasks demonstrate the method's promising potential. While I still have some reservations about the method's underlying intuition, I have raised my score to weak accept.

**Key Questions For Authors:**

1. Do you have preliminary results for Q2D2 on music/environmental audio, and what grid/hyperparameter tweaks would adapt its 2D quantization to non-speech feature correlations? Positive generalization evidence would elevate its assessment as a universal audio scheme; a lack would solidify its speech-only niche.
2. How does Q2D2 perform at medium/high bitrates (16/32 kbps+), and does its grid design suffer from codebook inefficiency with higher quantization levels? Sustained performance would position it as a full-bitrate alternative to VQ/RVQ; degradation would limit it to low-bitrate use cases.
3. How do Q2D2’s tokens perform on downstream generative tasks (TTS/voice conversion) vs. VQ/RVQ/FSQ? Strong generative results would demonstrate its value for end-to-end audio AI pipelines, expanding its assessed broader impact.

**Limitations:**

No. The authors only briefly note the speech-only focus (music/general audio as future work) and a generic ethical note on generative audio in the Impact Statement, with no substantive discussion of technical, deployment, or societal limitations.
Constructive Improvements:
explain why Q2D2’s 2D grid may fail for music/environmental audio (e.g., unstructured features, strong temporal correlations) and outline open adaptation questions.

**Strengths And Weaknesses:**

Soundness
Strengths: Technically rigorous with a reproducible Q2D2 pipeline and valid mathematical formulations; well-designed experiments with controlled baselines, diverse metrics and exhaustive ablations validate core design choices.
Weaknesses: No theoretical analysis of grid packing efficiency (only empirical validation); untested on music/general audio and downstream tasks like TTS; no high-bitrate scalability evaluation, leaving generalization questions unaddressed.
Presentation
Strengths: Clear structure and cohesive narrative with effective visualizations; thorough related work contextualization that distinguishes Q2D2 from PQ/FSQ/VQ; detailed pseudocode and hyperparameters ensure reproducibility.
Weaknesses: Redundant related work details and inconsistent mathematical notation; minimal inference cost comparison; weak links between ablation results and core contributions.
Significance
Strengths: Addresses a core audio quantization tradeoff (stability vs. feature correlation capture) with direct practical utility for low-bitrate speech communication/edge devices; opens a new geometry-aware quantization research direction.
Weaknesses: Impact is speech-specific (untested on universal audio); no industrial scalability evaluation or high-bitrate improvements; limited theoretical contribution to broader quantization research.
Originality
Strengths: First geometry-aware 2D quantization for audio, creatively combining grid-based quantization and implicit codebooks with audio-specific optimizations; provides novel empirical insights into geometric quantization for feature correlation capture.
Weaknesses: No entirely new theoretical framework; overlaps with product quantization without comprehensive comparison.

---

> ### Author Rebuttal · Authors · 2026-03-29
>
> We thank the reviewer for the constructive feedback and hope that our responses have addressed the concerns.
>
> ***Packing efficiency.** While our paper focuses on empirical validation, the advantage of the rhombic grid can be understood geometrically via nearest-neighbor distances, which directly affect quantization error. For a grid with spacings $d_x, d_y$:
>
> Rectangular:
> $$
> d_{\text{rect}} = \min(d_x, d_y)
> $$
>
> Hexagonal:
> $$
> d_{\text{hex}} = d_x, \quad d_y = \tfrac{\sqrt{3}}{2} d_x
> $$
>
> Rhombic:
> $$
> d_{\text{rhomb}} = \frac{\sqrt{d_x^2 + d_y^2}}{2}
> $$
>
> The rhombic lattice introduces diagonal neighbors, yielding smaller and more isotropic distances between points. This leads to denser local coverage and reduced quantization error for a fixed number of levels. This intuition aligns with our empirical results (Table 8), where rhombic grids consistently outperform rectangular and hexagonal ones.
>
> **Inference cost.** We analyze inference cost in terms of runtime, complexity, and memory. As shown in Table 13, Q2D2 achieves comparable RTF to WavTokenizer (0.0039 vs. 0.0032), indicating minimal overhead. Across quantization settings, memory remains stable (≈820–830 MB), and latency stays low (RTF 0.0024–0.025), demonstrating efficient inference across operating points.
> We agree this comparison was not sufficiently emphasized and will highlight it more clearly in the revision.
>
> **Q2D2 evaluation on music and environmental audio.** Q2D2 is trained on a large-scale (~8k hours) dataset spanning speech, audio, and music. Following the reviewer’s request, we evaluate it on the Arch benchmark. While the original submission (Table 7) focused on speech, here we include audio and music tasks: ESC-50, US8K, FSD50K, and VIVAE (acoustic events), and MTT, MS-DB (music).
>
> | Model   | Nq ↓ | token/s ↓ | MTT ↑ | MS-DB ↑ | ESC50 ↑ | US8K ↑ | FSD50K ↑ | VIVAE ↑ |
> |---------|------|------------|--------|----------|----------|---------|-----------|----------|
> | DAC     | 9    | 900        | 0.2805 | **0.6014**   | **0.2594**   | 0.4032  | 0.1297    | 0.3440   |
> | Encodec | 8    | 600        | 0.1993 | 0.3917   | 0.1925   | 0.3055  | 0.1091    | 0.3005   |
> | DAC     | 4    | 400        | 0.2784 | 0.5942   | 0.2580   | 0.3824  | 0.1293    | 0.3342   |
> | Encodec | 4    | 300        | 0.1934 | 0.3656   | 0.1790   | 0.3097  | 0.1099    | 0.2710   |
> | Encodec | 2    | 150        | 0.1900 | 0.3245   | 0.1699   | 0.2960  | 0.1065    | 0.2630   |
> | DAC     | 1    | 100        | 0.2502 | 0.5137   | 0.2065   | 0.3350  | 0.1295    | 0.2991   |
> | WavTokenizer| 1    | 75| 0.2835| 0.5764   | 0.2550   | 0.3975  | **0.1392**    | **0.3563**   |.
> | Q2D2| 1    | 53| **0.2847** | 0.5690   | 0.2567   | **0.4241**  | 0.1303    | 0.3365   |
>
> Q2D2 generalizes well beyond speech without additional tuning. It achieves the best results on MTT (music) and US8K (environmental audio), and remains competitive across all other datasets compared to strong baselines such as DAC and WavTokenizer.
>
>
> **Medium/high bitrates.** We evaluate Q2D2 across sampling rates (16, 24, 48 kHz), corresponding to increasing token rates (i.e., higher bitrates), while keeping all hyperparameters fixed. Results on LibriTTS test-clean are:
>
> | Model | SR   | token/s ↓ | UTMOS ↑ | PESQ ↑ | STOI ↑ | V/UV F1 ↑ |
> |-------|------|------------|----------|---------|---------|------------|
> | Q2D2  | 16000 | 111        | 3.9702   | 2.3441  | 0.9089  | 0.9378     |
> | Q2D2  | 24000 | 166        | 4.0312   | 2.3995  | 0.9152  | 0.9395     |
> | Q2D2  | 48000 | 333        | 3.9674   | 2.7190  | 0.9318  | 0.9498     |
>
> Q2D2 maintains strong and stable quality as bitrate increases, with consistent gains in PESQ, STOI, and V/UV F1. UTMOS remains relatively stable.  As a structured, codebook-free method, increasing tokens directly improves resolution without under-utilization issues. These results indicate that Q2D2 scales effectively beyond low-bitrate regimes and remains competitive at medium and high bitrates.
>
>
> **Downstream tasks.** Evaluating Q2D2 on generative tasks such as TTS is indeed important. We are currently running these experiments, but they are not yet complete and thus were not included in this submission. We aim to share results during the rebuttal period if available.
>
>
> **Theoretical contribution.** We agree that our work does not introduce a formal theoretical framework. Instead, Q2D2 provides a geometric formulation that bridges scalar (FSQ) and vector-based (VQ) methods via structured low-dimensional, codebook-free quantization. Formal analysis (e.g., linking geometry to error bounds) is left for future work and will be clarified in the revision.
>
> Regarding PQ, PQ uses learned codebooks over independent subspaces, while Q2D2 relies on fixed geometric lattices, inducing different Voronoi structures and avoiding codebook-related issues. While both partition dimensions, their mechanisms differ. We will aim to include a comparison to PQ in the rebuttal.

---

> > ### Author Rebuttal · Reviewer_D5nd · 2026-04-03
> >
> > Thank you for the author's feedback, my concens have been addressed.

---

> > > ### Author Response · Authors · 2026-04-03
> > >
> > > Thank you very much for your acknowledgment and for taking the time to review our rebuttal. We are glad to hear that our responses have addressed your concerns.
> > >
> > > If you feel it is appropriate, we would greatly appreciate it if you could consider updating your score to reflect your current assessment. Additionally, please let us know if there is anything further we can clarify or provide that could help in your evaluation, we would be happy to assist.
> > >
> > > Thank you again for your time and thoughtful feedback.

---

### Official Review · Reviewer_1fck · 2026-03-11

**Soundness:** 3
**Presentation:** 3
**Significance:** 2
**Originality:** 3
**Overall Recommendation:** 4
**Confidence:** 4

**Summary:**

This paper proposes Two-Dimensional Quantization (Q2D2), a geometry-aware quantizer for neural audio codec that group latent into 2 channels per group and quantizes each group on a fixed 2D tiling grid (hexagonal / rectangular / rhombic). The key goal is to keep FSQ-like stability and high utilization while capturing inter-channel correlations better than per-dimension scalar quantization. Experiments show that Q2D2 achieves state-of-the-art or competitive reconstruction quality at low token rates with high codebook utilization, without VQ-style auxiliary tricks.

**Compliance With Llm Reviewing Policy:**

Affirmed.

**Final Justification:**

The rebuttal addressed my main concerns, changed my evaluation

**Key Questions For Authors:**

When discussing and analyzing the 2D grid partitioning, it could be helpful to include visualizations of the Voronoi diagram, so readers can more intuitively understand the characteristics of different partitions and why the rhombic grid is a better choice.

**Limitations:**

No discussions on limitations. Maybe the authors can discuss more on interaction with LLM token modeling.

**Strengths And Weaknesses:**

Strengths:

(1) The idea is concise and easy to understand, and the technical approach is reasonable.

(2) The paper provides some in-depth analysis and discussion of the proposed algorithm, especially in the appendix where the discussion of the proposed 2D grid partitioning is fairly thorough. The corresponding ablation studies are also comprehensive.

Weaknesses:

(1) Concerns about the experimental setup:
- Why are Tables 2, 3, and 4 trained on different datasets?
- One observation: Table 3 uses the least data (585 hours), Table 4 uses the second most (8k hours), and Table 2 uses the most (150k hours). From the results, it appears that as the amount of training data increases, the advantage of the proposed method over other baselines becomes much weaker (i.e., Table 3 ≥ Table 4 > Table 2). Does this, to some extent, suggest that when the data scale is very large, the proposed method yields only limited performance gains?

(2) From my understanding, the core technical contribution of R2D2 can be summarized as “Group FSQ with 2 channels per group.” A natural follow-up question is: Which one performs better, Group FSQ with 2 channels per group, or Group VQ with 2 channels per group? The experiments in the paper mostly compare against FSQ and RVQ, but lacks a comparison with Group VQ under the same number of groups.

---

> ### Author Rebuttal · Authors · 2026-03-29
>
> Thank you very much for your review, we think we address all your concerns and hope that you will raise your rating.
>
> **Datasets, different inference tables and results** - We thank the reviewer for the observation. The use of different datasets across Tables 2–4 is due to computational constraints. Our goal was to compare against strong baselines as reported in prior work. However, retraining all methods at large scale (e.g., 150k hours) was not feasible. Therefore, we rely on pretrained models and reported results from the literature, which naturally involve different training data.
>
> Additionally, the tables evaluate performance under different conditions: Table 2 reports results on LibriSpeech test-clean, while Table 4 includes both clean and noisy datasets. Table 3 provides a fully controlled comparison, where all methods are trained by us under the same setting. In particular, this table isolates the comparison against FSQ and WavTokenizer, the framework on which Q2D2 is built.
>
> Regarding scaling and the score related to it, we believe that the observed differences are not directly driven by dataset size, but rather by dataset composition and diversity. Specifically, Tables 2 and 3 use datasets primarily from the speech domain (Table 2: Emilia + MLS and Table 3: LibriTTS), whereas Table 4 is trained on a more diverse mixture of speech, audio, and music data. This increased diversity introduces a broader range of signal characteristics, which can reduce the relative advantage observed in more homogeneous (speech-only) settings. Supporting this, we observe that when evaluating Q2D2 under comparable conditions (e.g., Table 2 vs. Table 3 on LibriSpeech test-clean with similar token rates of 66/75 tokens/s), the performance is nearly identical (UTMOS ≈ 4.04, PESQ ≈ 2.50, STOI ≈ 0.92), despite the large difference in dataset size.
>
> These results suggest that the variation in performance is more closely related to data diversity rather than scale alone, and that Q2D2 maintains consistent performance when evaluated under matched conditions.
>
> **Group FSQ and Group VQ** - While Q2D2 may appear similar to “Group FSQ with 2 channels per group,” the key distinction is that Q2D2 performs structured 2D quantization using geometric lattices (e.g., hexagonal, rhombic), rather than independent scalar quantization within each group. In FSQ, each dimension is quantized independently (equivalent to a hypercube partition), whereas Q2D2 jointly quantizes feature pairs, inducing non-axis-aligned Voronoi regions. This allows Q2D2 to capture inter-dimensional correlations that FSQ cannot model.
>
> Regarding Group VQ, it relies on learned codebooks, which introduce additional memory overhead, increased computational cost, and potential training instability (e.g., codebook collapse). In contrast, Q2D2 is codebook-free, lightweight, and fully deterministic, while still modeling dependencies between feature dimensions through its structured partitions.
>
> We agree that a direct comparison with Group VQ under the same grouping would be valuable. We will try to conduct this evaluation during the rebattal. However, prior work suggests that VQ-based methods typically require larger codebooks and higher complexity to achieve similar efficiency, particularly in low-bitrate regimes.
>
> **Voronoi Diagram** - as suggested by the reviewer we will add  Voronoi diagram to the new version.
>
> **LLM token modeling and limitations** - As noted in the paper, evaluating Q2D2 in downstream generative tasks such as text-to-speech (TTS) is an important direction for future work. We are currently conducting these evaluations. However, they are not yet complete and therefore were not included in this submission. We hope it will be ready for the rebuttal period.
>
> We believe that Q2D2 will have strong performance in generative settings that will further demonstrate the value for end-to-end audio modeling, and we will include these results in a future revision.

---

> > ### Author Rebuttal · Reviewer_1fck · 2026-04-02
> >
> > Raised score as concerns have been addressed.

---

### Decision · Program_Chairs · 2026-04-30

**Decision:**

Accept (regular)

**Comment:**

In the initial review, the major concerns were about the lack of validation in downstream generation tasks like TTS and the absence of comparisons with Group VQ. The rebuttal convinced the majority of the reviewers by providing solid subjective CMOS results for TTS. While one reviewer has doubts about the efficiency of the 2D grid quantization at exceptionally high bitrates, the overall strength at practical bitrates outweighs the weakness. Thus, the AC leans towards acceptance.